# High-resolution separation of bioisomers using ion cloud profiling

Xiaoyu Zhou [1,2], Zhuofan Wang[1], Jingjin Fan[1] & Zheng Ouyang [1,2] ✉

Elucidation of complex structures of biomolecules plays a key role in the field of chemistry and life sciences. In the past decade, ion mobility, by coupling with mass spectrometry, has become a unique tool for distinguishing isomers and isoforms of biomolecules. In this study, we develop a concept for performing ion mobility analysis using an ion trap, which enables isomer separation under ultra-high fields to achieve super high resolutions over 10,000. The potential of this technology has been demonstrated for analysis of isomers for biomolecules including disaccharides, phospholipids, and peptides with post-translational modifications.

Biomolecules, such as glycans, lipids, and peptides, play vital roles in biological systems[1–3]. They exist in forms of a wide variety of isomers or isoforms, which have identical chemical formulas and molecular weights but different structures and biological functions[4]. The glycans[5–9] and lipids[10–12] have complex configurations, while the structural characterization of peptides and proteins can be complicated with post-translational modifications (PTMs) and higher-order conformations, all resulting in a variety of isomers and isoforms[13–15]. The significance of identifying isomers is well recognized for almost all disciplinaries[16]. As one example, isomers such as cis or trans unsaturated fatty acids in food can have very different effects on human health[17].

Ionic forms of the molecules have been used as surrogates for species identification as well as structural characterization. While mass spectrometry (MS) is often used to obtain the compositions of molecular ions[18–21], ion mobility (IM) has been widely employed for differentiation of the isomers and isoforms of biomolecular ions[22–24]. The combination of these two technologies in the form of IM-MS has been a major direction in mass spectrometry development during the last decade. IM techniques separate ions based on the mobility differences due to the collisions between the ions and the background neutral molecules. Taking the most classic method with drift-tube ion mobility as an example, it uses ion-neutral collisions in an electric field (with a field strength $E$) to identify the differences in collision cross-section (CCS), which is associated with the variations in structures of molecular ions. Applications of IM have led to the separation of the isomers or isoforms for disaccharides, phospholipids, and peptides with post-translational modifications[25–27]. The separation efficiency is affected by both the electric field $E$ and the number density ($N$) of the collision gas. The mobility of the ions is the result of the balance between the motion driven by the electric field E and the collisions with the background gas molecules. Typically, IM is performed under a low electric field $E/N < 30$ Td (Townsend number, $1\,\mathrm{Td} = 1 \times 10^{-21}$ V m²)[28], such as for implementations with drift time[22], traveling wave[29], trapped[30, 31], and differential modes[32]. With a long separation time[30] or path[33–36], separation resolutions of several hundreds have been achieved (Supplementary Fig. 1 and Supplementary Table 1), which, however, is still inadequate for differentiating bioisomers of high structural complexities.

Operating the IM at higher $E/N$ ratios, for example, applying high-field asymmetric waveform at ~100 Td, shows enhanced isomer separation;[37] however, researchers have not discovered an effective method for implementation with higher $E/N$ ratios. The main challenge is to prevent the loss of the ions while they are driven by a high electric field through the collisions over a relatively long time or long path.

Here, we report the development of ion cloud profiling technology to enable high-resolution IM analysis in a radiofrequency (RF) electromagnetic field with $E/N > 1$ MTd. Isomer differentiation was achieved at a resolution better than 10,000, which allowed us to attack some challenging issues in the structural analysis of biomolecules.

## Results and discussion

### Ion cloud profiling: instrumentation and performance

The ion cloud profiling was performed in a dual-LIT (linear ion trap) miniature mass spectrometer modified from a Mini β instrument

[1]State Key Laboratory of Precision Measurement Technology and Instruments, Department of Precision Instrument, Tsinghua University, Beijing 100084, China. [2]Institute for Precision Medicine, Tsinghua University, Beijing 100084, China. ✉e-mail: ouyang@tsinghua.edu.cn

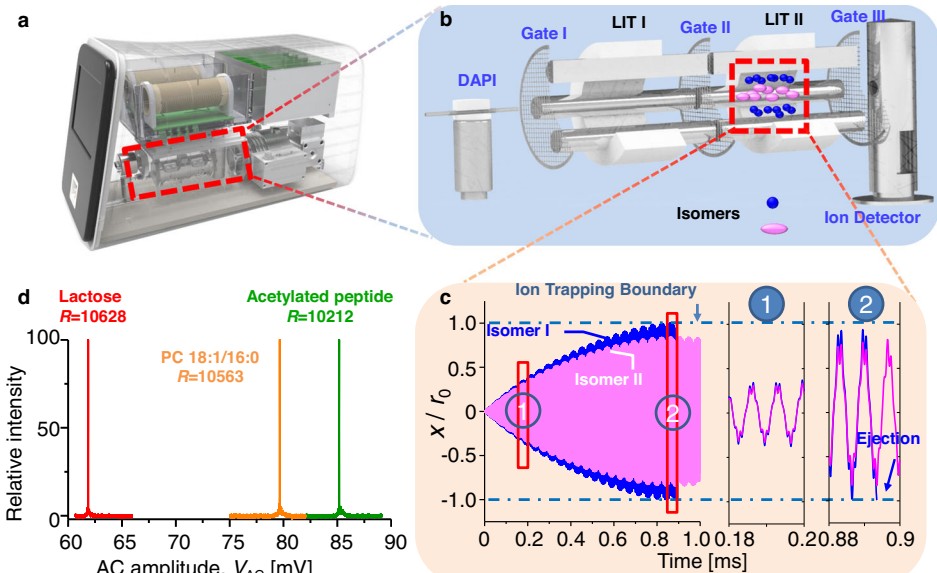

**Fig. 1 | Instrumental setup, principal, and performance characterization of the ion cloud profiling technology.** Schematics of **a** the miniature MS system used in this work and **b** its key components for isomer structural analysis. **c** Simulated ion trajectories for two isomeric ions characterized by reduced damping coefficients: $b' = 0.0010$ (blue) and 0.0012 (purple). Here, $b' = 2b/\Omega m$, where $\Omega$ is the angular frequence of the RF field, $m$ is ion mass, $b$ is the damping coefficient of the ions. Insets, zoom-in plots of the ion trajectories at the beginning (1) and ejection (2) stages of the AC excitation. Isomeric ions are ejected sequentially according to their DCSs when their oscillation amplitudes exceed the trap geometry, $r_0$, as indicated by the blue dashed lines. **d** Ion cloud profiling spectra of three biomolecules, lactose ($m/z$ 365, CCS 177.6 Å$^2$), phosphatidylcholine (PC) 18:1/16:0 ($m/z$ 761, CCS 296.2 Å$^2$), and an acetylated peptide ($m/z$ 542, CCS 357.9 Å$^2$), superimposed in one spectrum. The CCS values of lactose and peptide are measured by timsTOF (Bruker Daltonics, Bremen, Germany). The CCS value of phosphatidylcholine is taken from *Groessl's* work[42]. The resolution here is defined as, $R = V_{AC}/\triangle V_{AC}$, where $V_{AC}$ and $\triangle V_{AC}$ are the AC ejection amplitude of analyte ions and the full width at half maximum (FWHM) of the peak, respectively[43].

(PURSPEC Technology (Beijing) Ltd., Beijing China) (Fig. 1a), which is also capable of performing tandem MS analysis[38]. Ions of isomers were generated by a nano-electrospray ionization (nESI) source, mass selected in the LIT I, and transferred to the LIT II for final structural analysis (Fig. 1b). The physics employed here was that under forced oscillations, ion clouds of the isomeric ions were separated due to their difference of damping cross-sections (DCSs, Fig. 1c)[39–41]. For experimental implementations, an auxiliary alternative-current (AC) was employed for resonance excitation of the isomeric ions in the LIT; by scanning the AC voltages in different scan rates depending on the species, the isomeric ions with different DCSs were ejected sequentially according to the ion cloud profile sizes (Fig. 1c and Supplementary Fig. 2). When the AC scanning speed was below 5000 mV/s, a good correlation between AC ejection voltages and DCSs could be established and an ion cloud profiling spectrum with a resolution over 10000 (Fig. 1d) was produced. More specifically, the AC scanning speed was 53 mV/s for the analysis of disaccharides, and 88 mV/s for both phospholipids and peptides (Fig. 1d).

## Analysis of biomolecules

To show the enhanced separation capability of the technology, we first analyzed glycans with complexed isomeric structures. Four disaccharides, including trehalose, maltose, cellose, and lactose, were considered. They present differences in the structure include composition, connectivity, and configuration (Fig. 2a). The disaccharides are composed of basic building blocks, the monosaccharides. Each monosaccharide contains multiple hydroxyl groups, which could be connected to form a glycosidic bond with another monosaccharide. Through linking different hydroxyl groups, carbohydrates normally have branched structures with diverse regiochemistry. In addition, each glycosidic bond formation is accompanied by creation of a stereocenter, because two monosaccharides could be connected via two different configurations. Using ion cloud profiling method developed here, the four isomeric disaccharides were baseline resolved (Fig. 2b)

and the experimental results agreed well with the simulation (Supplementary Fig. 3). A mixture of lactose and cellose was also analyzed, by both IM through ion cloud profiling (Fig. 2c) and MS/MS analysis (Top, Fig. 2d). While MS/MS normally is powerful for distinguish isomers through characteristic fragmentation patterns, in this case identical patterns were observed for these two isomers. The results here suggest that the IM using ion cloud profiling technology is complementary to the MS or MS/MS analysis and has the potential to achieve high-resolution structural analysis of isomers, leading to isomeric-specific selection (Supplementary Fig. 4) and quantification analysis (Figs. 2e and 2f).

To show the universality of the technology, we further demonstrated the structural separation of phospholipids (Fig. 3a) and peptides (Fig. 3e). Phospholipids consist of two fatty acyl chains, a phosphate head group, and a glycerol backbone, for which the isomerization arises from a number of variants in structure including sn-positions of the fatty acyl chains (Fig. 3b), C=C locations (Fig. 3c) and configurations (cis/trans, Fig. 3d) of C=C in each fatty acyl chain, derivatization sites, and etc. These structure features could be well characterized using the ion cloud profiling (Fig. 3b–d). For proteins and peptides, PTMs contribute significantly to the structure complexity. In this study, peptides SGKLRASHKG with methylation in K3 or K9 (Fig. 3f), acetylation in K3 or K9 (Fig. 3g), and phosphorylation in S1 or S7 (Fig. 3h) were analyzed for demonstration. As shown in Fig. 3f–h, the isomers of the peptides could all be clearly resolved in the ion cloud profiling spectra. This method has also been applied for distinguishing different conformations of protein ions, for which baseline separation was obtained for charge states of +9 and +13 of myoglobin as well as +12 and 13 for cyochrome c (Supplementary Fig. 5).

In summary, we show an ion mobility technology for high-resolution structural separation of bioisomers. The DCS-based ion cloud profiling could serve as an alternative means for structural separation of biomolecules. The technology developed here showed distinct advantages with a resolution over 10,000, which represents a

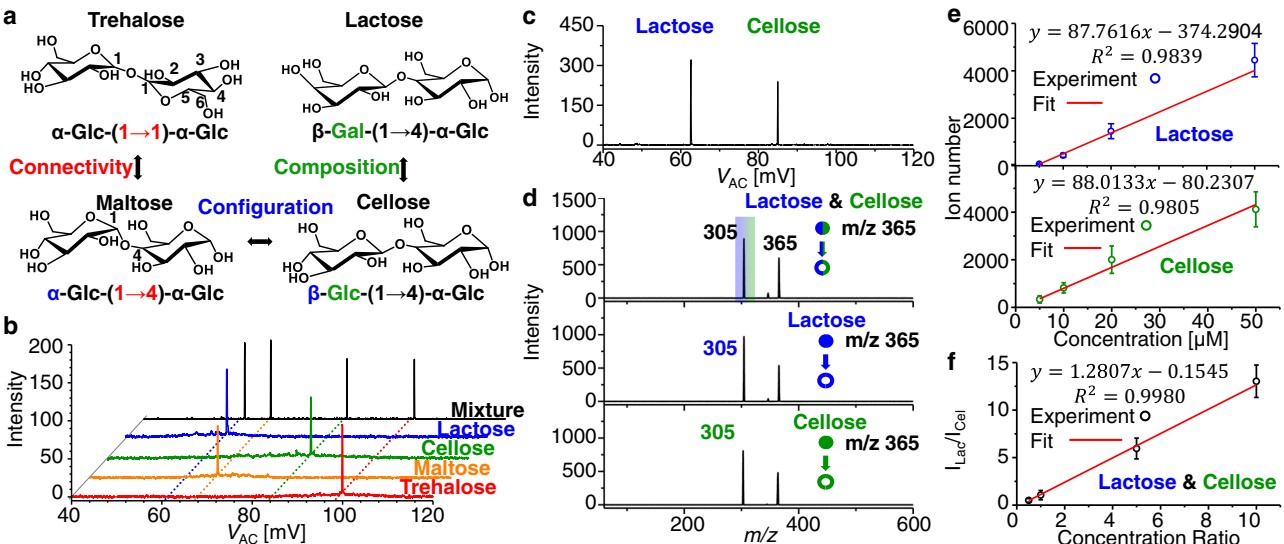

**Fig. 2 | Structural analysis of glycans. a** Structures of four isomeric disaccharides, which have isomerization of composition, connectivity, and configuration in pairs. **b** Ion cloud profiling spectra of the four disaccharides: trehalose (red), maltose (orange), cellose (green), lactose (blue), and the mixture of these four (black). **c** Ion cloud profiling spectrum of lactose and cellose mixture. **d** Tandem MS spectra of lactose and cellose mixture (top), pure lactose (middle), and pure trehalose (bottom). Lactose (blue) and cellose (green) have identical mass to charge ratio, m/z 365, and fragment, m/z 305, in tandem MS spectra. **e** Calibration curves for pure lactose and cellose. For quantitative analysis of pure sample, concentrations varied from 5 μM to 50 μM. Each value represents the mean ± s.d. ($N = 10$). **f** Calibration curves for the mixture of lactose and cellose. For quantitative analysis of mixture, concentration of cellose was 5 μM, and concentration ratios of lactose to cellose varied from 0.5 to 10. Each value represents the mean ± s.d. ($N = 15$). Source data are provided as a Source Data file.

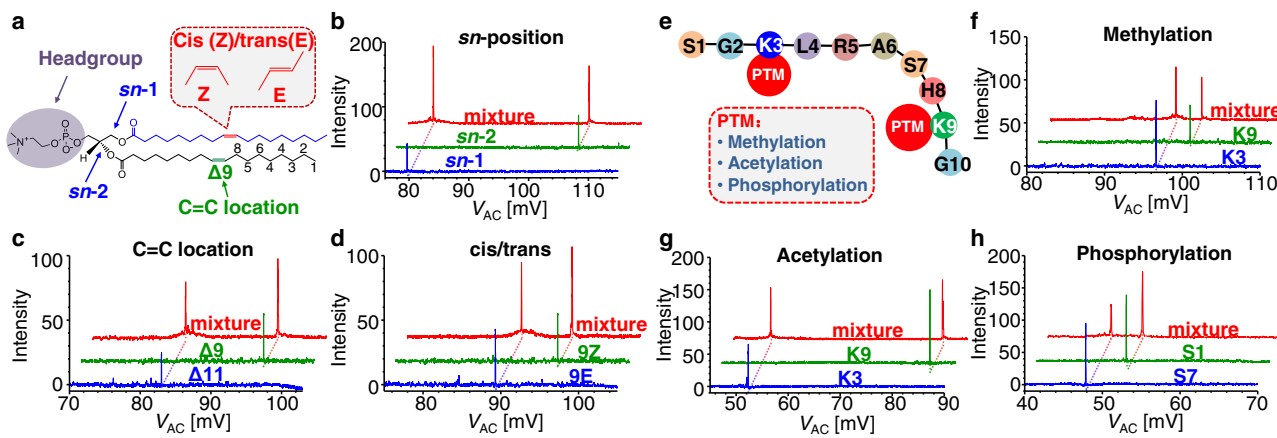

**Fig. 3 | Structural analysis of lipids and peptides. a** Structure of phospholipids with isomerization of sn position, carbon-carbon double bond location, and cis/trans structure due to the double bond. Ion cloud profiling spectra of **b** PC 18:1(9Z)/16:0 (blue), PC 16:0/18:1(9Z) (green), and their mixture (red); **c** PC 18:1 (11Z)/18:1(11Z) (blue), PC 18:1 (9Z)/18:1(9Z) (green), and their mixture (red); **d** PC 18:1 (9E)/18:1(9E) (blue), PC 18:1 (9Z)/18:1(9Z) (green), and their mixture (red). **e** Structure of peptide, SGKLRASHKG, with different types of PTMs. Ion cloud profiling spectra of the peptide with **f** methylation in K3 (blue), K9 (green), and their mixture (red); **g** acetylation in K3 (blue), K9 (green), and their mixture (red); **h** phosphorylation in S1 (blue), S7 (green), and their mixture (red).

significant improvement in analytical technology for biological studies (Supplementary Fig. 1). Moreover, the implementation is very simple with the use of an ion trap, which can perform MS/MS analysis at the same time. Ion trap also is a popular ion processing device in modern hybrid mass spectrometers and the high $E/N$ IM separation performed at low pressure make it highly compatible with coupling to mass analyzers such as Orbitrap and TOF. It is expected that this method can be readily applied for a broad range of applications for biological study.

## Methods

### Materials

Trehalose (200 μM) was purchased from Klamar reagent (Shanghai, China), maltose (200 μM) was purchased from Meryer (Shanghai, China), cellose (200 μM) was purchased from Macklin (Shanghai, China), lactose (200 μM) was purchased from Aladdin (Shanghai, China). Synthetic lipids standards, PC 16:0/18:1(9Z) (200 μM), PC 18:0/16:1(9Z) (200 μM), PC 18:1/18:1(9Z) (200 μM), PC 18:1/18:1(9E) (200 μM) and PC 18:1/18:1(11Z) (200 μM), were purchased from Avanti Polar Lipids (Alabaster, AL, USA). Six kinds of peptides (50 μM), SGKLRASHKG with phosphorylation in S1 and S7, methylation in K3 and K9, acetylation in K3 and K9, were synthesized and purchased from Sangon Biotech (Shanghai, China). These samples were used directly without further purification. Methanol and water, purchased from Fisher Scientific (Fairlawn, NJ, USA) were used for preparing the sample solvents. Trehalose, maltose, cellose, lactose and six peptides were solvated in methanol and water (50/50, v/v). Lipids standards were solvated in methanol with 0.1% acetic acid.

## MS instrumentation

The experiments were performed in a home-made dual-linear ion trap (dual-LIT) miniature MS system (Fig. 1a)[38], which includes a nano-electrospray (nESI) ion source for sample ionization in atmosphere, a discontinuous atmospheric pressure interface (DAPI) connecting the atmosphere and the vacuum, and dual-LIT mass analyzers, LIT I and LIT II, as well as three gates for ion processing. The DAPI uses a pinch valve to control the ion introduction during each analysis. The DAPI typically opens 5-30 ms to allow the ion introduction and then closes to allow the pressure drop back from 0.1 Torr (13.3 Pa) to $1 \times 10^{-5}$ Torr. Each of the LITs had a nominal radius $r_0$ of 4 mm, and a length $z_0$ of 51 mm, driven by a dual-phase RF at a frequency of 1 MHz for ion trapping and mass analysis. The two LITs were separated by three mesh gates, which were applied with direct-current (DC) voltages to tune the ion transfer along the $z$ direction (Table S2). With a small alternating-current (AC) voltages coupled with the RF, resonance excitation of the ions in the LITs was achieved to allow ion isolation, ion activation, and ion ejection (Table S3). Air was used as the buffer for ion cooling. For IM technique, typically the detector response speed is not the limiting factor for the resolution and the instrument operation status is not approaching the detector response speed limit. All data were collected and processed by SpecMS from PURSPEC. All data processing were performed using MATLAB 2017b from MathWorks.

## Theoretical modeling and numeric simulations

Theoretical modeling and numeric simulations were employed for the understanding and optimization of the profiling process. When the ions were excited by an AC, the motion of the ions in the LIT II was described by a forced oscillation. The equation yields:

$$m\frac{d^2x}{dt^2} + b\frac{dx}{dt} + kx = C\sin(\omega t) \tag{1}$$

where $m$ is ion mass, $x$ is ion displacement in the $x$-coordinate, $t$ is time. $C$ represents the excitation strength and is defined as

$$C = \alpha V_{AC}/2r_0 \tag{2}$$

where $V_{AC}$ is AC voltage with an angular frequency $\omega$ and $\alpha$ is calibration coefficient. $k$ is the spring constant due to the effective RF field and is defined as

$$k = 2eV_{eff}/r_0^2 \tag{3}$$

where $V_{eff}$ is the effective trapping depth of the RF field, $e$ is electron charge, and $r_0$ is the field radius of the LIT II. $b$ is the damping coefficient of the ion motion and is defined as

$$b = eK^{-1} \tag{4}$$

$K$ is the ion mobility in the RF field and is defined as

$$K = \frac{3e}{16N}\sqrt{\frac{2\pi}{\mu k_B T}}\frac{1}{\Omega_D} \tag{5}$$

where $\Omega_D$ is the DCS of the ions, $k_B$ is Boltzmann constant, and $\mu$ is the reduced mass of ions and collision gas, $T$ is temperature.

Therefore, the damping coefficient $b$ is proportional to the damping cross-section from Eq. 4, and Eq. 5:

$$b = \frac{16N}{3}\sqrt{\frac{\mu k_B T}{2\pi}}\Omega_D \tag{6}$$

The solution of Eq. 1 yields:

$$x = \gamma e^{-\frac{b}{2m}t}\sin\left(\sqrt{\omega_0^2 - \frac{b^2}{4m^2}}t + \delta\right) + \left[\frac{(\omega_0^2 - \omega^2)\sin(\omega t) - \frac{b\omega}{m}\cos(\omega t)}{(\omega_0^2 - \omega^2)^2 + \left(\frac{b\omega}{m}\right)^2}\right]\frac{C}{m} \tag{7}$$

where $\omega_0^2 = \frac{k}{m}$, $\gamma$ and $\delta$ are arbitrary constant depending on the initial conditions of the ions. The first term represents a damped ion motion, whose oscillation amplitude goes to zero within a few milliseconds. The second term represent the forced oscillation of the ions. At resonance, i.e., $\omega = \omega_0$ (resonance frequency), the maximum displacement of the ion motion, $A$, as a function of the damping term, $b$, yields:

$$A = \frac{C}{b\omega_0} = \sigma b^{-1} \propto \Omega_D^{-1} \tag{8}$$

where $\sigma = C\sqrt{\frac{m}{k}}$ is a constant for a specific ion trapping condition.

Equation 8 is the theoretical basis of the ion cloud profiling technology for performing structural analysis of isomeric ions. Under the same AC excitation, the size of the ion cloud, $A$, becomes different for isomers according to the damping term $b$ or DCS $\Omega_D$, as shown in Fig. 1c.

Especially, if the displacement of the ion motion is equal to $r_0$, we can obtain the correlation between scanning AC voltage and DCS $\Omega_D$, which has

$$V_{AC} = \frac{32N}{3}\sqrt{\frac{\mu k_B T}{2\pi}}\alpha^{-1}r_0^2\omega_0\Omega_D \tag{9}$$

The DCS of ions is proportional to the AC ejection voltage. And the larger the DCS is, the greater the AC ejection voltage is required.

To obtain the high $E/N$ condition, the pressure for the analysis was set below $1 \times 10^{-5}$ Torr. Adequate separation of isomers could be achieved, while the ion pack being maintained compact for each species (Supplementary Fig. 2) to allow high resolutions to be achieved through the ion cloud profiling method. For experimental implementation, high-resolution spectra were obtained by profiling the isomeric ion clouds at conditions optimized (Fig. 1d, Supplementary Figs. 6–10).

### Reporting summary

Further information on research design is available in the Nature Portfolio Reporting Summary linked to this article.

## Data availability

All data supporting the findings of this study are available from the corresponding authors upon request. The raw data are available from Figshare (https://doi.org/10.6084/m9.figshare.22139816.v1). Source data are provided with this paper.

## Code availability

The codes used for data acquisition and processing are available from the corresponding authors upon request. The codes used for simulation are available from Github (https://github.com/isShuaiLi/QIT.git).

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

## Acknowledgements

We thank professor Yu Xia, Xiaoxiao Ma, and Wenpeng Zhang at Tsinghua University for their helpful discussions. This work was supported by the National Natural Science Foundation (Project No. 21627807 (Z.O.) and 21934003 (Z.O.)) and Tsinghua University Initiative Scientific Research Program of Precision Medicine. We also thank PURSPEC Technology (Beijing) Ltd. for support to instrument modification.

## Author contributions

X.Z. and Z.O. designed the research; X.Z., Z.W., and J.F. performed the investigation; X.Z. and Z.O. wrote and edited the manuscript.

## Competing interests

The authors declare no competing interests.
