## [Peer Review File · Nature Communications]

REVIEWER COMMENTS

Reviewer #1 (Remarks to the Author):

This work demonstrates the separation of isomers based on differential ion ejection from an ion trap based on ion cross-section. The authors show impressive resolution using the new method. The concept of size-based separations using an ion trap is compelling, interesting, and worthwhile for a large audience. Several revisions are recommended prior to publication:

- The background information on ion mobility methods is not entirely accurate. The authors state 'IM uses ion-neutral collisions in an electric field (with a field strength E) to identify the differences in collision cross-section (CCS) due to the variations in molecular structures...' which is true of drift tube ion mobility, but not of the many other IM methods that exist (and in fact which the authors showcase in their supplementary information.)
- The concept of E versus N and Townsend number are not familiar for most readers, and this concept needs more explanation.
- The authors mention a couple arbitrary examples of isomers in the first paragraph (glycans, lipids, peptides). Everyone knows the concept of isomers and the importance of differentiating them. This is common knowledge. Giving three specific examples is not relevant.
- Citing reference 16 as a general example of the use of mass spectrometry to examine molecular compositions is odd.
- In Figure 1, it is not clear why the damping coefficients for two isomeric ions have different number of significant digits. This seems like a significant oversight or merits discussion if it is a real difference (especially for a method that focuses on sophisticated analytical measurements).
- It is not clear how the damping cross-sections correlate with the damping coefficients.
- In Figure 1, resolutions on the order of 10,000 are reported, but the sizes (masses and cross-sections) of the three ions are not given.
- It is not clear how the AC excitation is applied and ramped in order to cause selective ejection of the three ions in Figure 1. What is the amplitude and gradient? These are the most important features/parameters and should be included in the main text (and cut the unnecessary common-knowledge information about lipid and glycan isomers). In general, there is key information included in Supporting that is not mentioned in the main text, and much of it is more important than some of the information that is included in the main text.
- Some of the paragraphs are padded with unnecessary facts that are common knowledge, such as "The disaccharides are composed of basic building blocks, the monosaccharides. Each monosaccharide contains multiple hydroxyl groups, which could be connected to form a glycosidic bond with another monosaccharide. Through linking different hydroxyl groups, carbohydrates

normally have branched structures with diverse regiochemistry. In addition, each glycosidic bond formation is accompanied by creation of a stereocenter, because two monosaccharides could be connected via two different configurations.” The same unnecessary padding occurs in the description of phospholipids and peptides.

- The term “profiling spectra” is too vague. Please consistently use the term “ion cloud profiling spectra”.
- It appears that the ion cloud profiling spectra do not retain quantitative information about the composition of the mixtures. Is this true or are the ion cloud profiling spectra not displayed in an accurate manner?
- It is suggested that the ion cloud profiling method is a new separation method. Are the ions truly separated, meaning that they could be captured as separate populations? Or are they simply detected as separated populations?
- <https://pubs.acs.org/doi/full/10.1007/s13361-017-1720-1> This paper is not cited which previously has shown that CCS information can be determined using an ion trap.
- The impact of the paper would be improved significantly if a correlation can be made between the scanning voltages that are applied and cross section of the species .
 - o Do the voltages used have any correlation to the actual size of the species? For Figure 3F and 3G: both modifications are on the same amino acid and have a similar sized modification- why is the Vac for the K3 acetylation so low in comparison to the Vac for K9 acetylation and K9/K3 methylation? There doesn't seem to be a distinct correlation of molecular size and Voltage (especially in comparison with Figure 2 displaying Vac for sugars)
- Figure S8 indicates that ion cloud profiling can be used to separate the isomers, select and fragment only one of the isomers, and then detect the unique fragmentation pattern. In the dual ion trap they state that ion cloud profiling is conducted in the second/back ion trap so it is not clear how the ions could be ejected from this trap to generate the separation but also fragment just one species and then detect it.
- Does the ‘resolution’ of the separation depend on the ion detector used? Would a faster detector offer higher resolution?
- All species studied within this work appear to be of single ‘conformation’. It would be worthwhile to show data where a typical mobiligram may show multiple peaks present and how this new ion cloud profiling may distinguish these peaks as opposed to just showing pure isomers and mixtures of them. Many small peptides have various conformations and would be sufficient for this work to prove that distinct features or conformations are observed with this method.
- They mention the potential for use with a TOF or Orbitrap but provide no further information. This seems speculative at best.

Reviewer #2 (Remarks to the Author):

This is a very interesting paper that reminds me of a 1999 paper from Graham Cooks that involved seeing “chemical shifts” for isomers analyzed by ion trap MS. The authors have introduced an interesting change of instrumental geometry (compared with Cooks), added off axis fields to turn it into a high resolution technique, and with these changes report a resolving power of 10,000. This is truly spectacular and is enabling for some types of systems – as the authors have demonstrated! The approach that they have taken would allow them to build a library isomers and by scanning in the extra dimension characterize at least some types (and perhaps many types) of isomers. This cannot be done by MS alone. I would not refer to this as a mobility technique; rather I believe they are taking advantage of collisional dampening, which also depends upon the shape. Whatever it is called, this approach will very likely be used broadly for many types of problems. It is important in the field of MS.

I do have some concerns and suggestions for improvement.

1) My take is that the authors have not resolved any pair of ions that cannot be done with existing technology. That is, they resolve species that I believe can be resolved by traditional IM-MS methods and these methods now have quite high resolving powers. See the SLIMS work, circular drift tube and tims work, all commercially available). I have not checked this carefully. But, it would be very straightforward for the authors to choose a system that we know has many structures that cannot be resolved. That is, does the resolving power of the technique allow them to resolve two different chemical species, which differ in structure by 0.01%. Showing that these can be resolved would give the technique a substantial advantage. Some of the reported spectra show a baseline that makes me suspect that these types of ions cannot be separated with this approach. That is, if closely related species were resolvable they should see peaks rather than a continuous baseline.

2) Another very exciting aspect of the work is associated with molecular conformation. I would expect the authors’ ions to show multiple structures; but, again they are not resolving any differences. This suggests that the dampening essentially anneals the ions to a common structure? Or perhaps the ions remain extremely hot and they have measured an average of all structures for very hot ions? They might add another ionizing couterion – perhaps Ca^{2+} to trap these; or, they could examine a system that is known to display different geometric configurations.

I believe this work will have a substantial impact wherever it is published. However, I’m not sure that it meets the level of uniqueness and includes a new discovery that are typically required to exceed the high threshold associated with Nature.

Reviewer #3 (Remarks to the Author):

Publish after Minor Revisions

The work from Ouyang and co-workers is of high rigor and novelty and will have major impact across analytical and life sciences. Their proposed ion mobility platform, using an easily accessible ion trap at high electric fields, represents a substantial progress in ion mobility instrument development that can be applied to separate isomers of biomolecules.

This said, in order for this work to be appropriate for the broad audience of Nature Communications, the impact needs to be better framed. I recommend publication after the following comments have been addressed:

Introduction (lines: 17 – 29): The existence of different isomers and isoforms have been described here and throughout the manuscript, perhaps in a slightly repetitive way. What is missing is a rationale for why the separation of isomers/isoforms is so crucial and what the implications are.

The readership of Nature Communications is broad and this is not necessarily obvious for readers from other fields. Here it would be helpful to describe a specific, and broadly accessible example with potential pharmaceutical implications, where commercial state of the art technology reached its limits. It is crucial to describe this impact more clearly in order for the paper to be suitable for this general journal.

Line 40: It would be helpful to add that these are normally “low” electric fields.

IM-MS directly measures the properties of molecular ions and not *just* molecules, please correct that to avoid any misunderstanding for readers not from the field (lines 32, 33, 36).

Figure 1: C) Suggest changing the color scheme for readability D) It would be preferred to see a spectrum of the two simulated ions described in B) and C), if possible also simulated. The panel that is now D) could go to E).

Figure S8: Typo in the caption, it is 347 instead of 307. Please comment on why the $m/z = 347$ peak is not visible in the MS2 spectrum of lactose. As it is presented now, the comparability is limited.

Figure S9: I suggest this, or something similar, should appear and be described in more detail in the main paper. The ability to quantify different isomers/isoforms of biomolecules in complex mixtures using IM-MS is of high impact. While the principle is illustrated in Figure S9, it would be good to extend this and e.g. quantify the precision with which the mixture can be analysed.

It would be helpful to have an overview table in the main paper, where key properties of your platform (type of IM separation, resolution, cost?, usability, CCS determination possible yes/no etc., similar to Table 1 in 10.1007/s00216-019-01807-0) are compared with both state of the art commercial instruments (please report the highest resolutions known so far for the instruments, e.g. different for Cyclic from the originally reported ones), but also possibly with home-built instruments. Is the reported ion mobility platform the one with the highest resolution so far? This is not clear from the manuscript and needs to be commented on.

Possible Missing Citations in the introduction.

10.1002/mas.21585

10.1007/s13361-019-02288-2

10.1007/s00216-019-01807-0

The SI Figures are not in the order as they appear in the manuscript.

There are a few typos and phrases where readability could be improved (lines 17, 37, 48, 78, 147).

Line 48, Here a new paragraph should start.

Response to reviewers' comments

Reviewer #1 (Remarks to the Author):

This work demonstrates the separation of isomers based on differential ion ejection from an ion trap based on ion cross-section. The authors show impressive resolution using the new method. The concept of size-based separations using an ion trap is compelling, interesting, and worthwhile for a large audience. Several revisions are recommended prior to publication:

- (1) The background information on ion mobility methods is not entirely accurate. The authors state 'IM uses ion-neutral collisions in an electric field (with a field strength E) to identify the differences in collision cross-section (CCS) due to the variations in molecular structures...' which is true of drift tube ion mobility, but not of the many other IM methods that exist (and in fact which the authors showcase in their supplementary information.)

Re: We have made revisions on Page 2 Line 38 to make the description more accurate:

"IM techniques separate ions based on the mobility differences due to the collisions between the ions and the background neutral molecules. Taking the most classic method with drift-tube ion mobility as an example, it uses ion-neutral collisions in an electric field (with a field strength E) to identify the differences in collision cross-section (CCS), which is associated with the variations in structures of molecular ions."

- (2) The concept of E versus N and Townsend number is not familiar to most readers, and this concept needs more explanation.

Re: For clarification, the description of E/N has been revised on Page 2 Line 37 as follows:

"The separation efficiency is affected by both the electric field E and the number density (N) of the collision gas. The mobility of the ions is the result of the balance between the motion driven by the electric field E and the collisions with the background gas molecules. Typically, IM is performed under a low electric field $E/N < 30$ Td (Townsend number, $1 \text{ Td} = 1 \times 10^{-21} \text{ V}\cdot\text{m}^{-2}$)"

- (3) The authors mention a couple of arbitrary examples of isomers in the first paragraph (glycans, lipids, peptides). Everyone knows the concept of isomers and the importance of differentiating them. This is common knowledge. Giving three specific examples is not relevant.

Re: We have revised the first paragraph as follows:

“The glycans ⁵⁻⁹ and lipids ¹⁰⁻¹² have complex configurations, while the structure characterization of peptides and proteins can be complicated with post-translational modifications (PTMs) and higher-order conformations, all resulting in a variety of isomers and isoforms ¹³⁻¹⁵. The significance in identifying isomers is well recognized for almost all disciplines ¹⁶. As one example, isomers such as cis or trans unsaturated fatty acids in food can have very different effects on human health ¹⁷.”

- (4) Citing reference 16 as a general example of the use of mass spectrometry to examine molecular compositions is odd.

Re: We have added three new references as follows:

- 19 Glish, G. L. & Vachet, R. W. The basics of mass spectrometry in the twenty-first century. *Nature Reviews Drug Discovery* **2**, 140-150 (2003).
- 20 McLuckey, S. A. & Wells, J. M. Mass analysis at the advent of the 21st century. *Chem. Rev.* **101**, 571-606 (2001).
- 21 Tamara, S., den Boer, M. A. & Heck, A. J. R. High-Resolution Native Mass Spectrometry. *Chem. Rev.* **122**, 7269-7326 (2022).

Please see Page 2 Line 28 of the manuscript:

“While mass spectrometry (MS) is often used to obtain the molecular compositions ¹⁸⁻²¹, ion mobility (IM) has been widely employed for differentiation of the isomers and isoforms of biomolecules ²²⁻²⁴.”

- (5) In Figure 1, it is not clear why the damping coefficients for two isomeric ions have the different number of significant digits. This seems like a significant oversight or merits

discussion if it is a real difference (especially for a method that focuses on sophisticated analytical measurements).

Re: The damping coefficient b in Fig. 1 was to represent typical values of two isomers, and we have corrected the number of significant digits to $b' = 0.0010$ and 0.0012 .

(6) It is not clear how the damping cross-sections correlate with the damping coefficients.

Re: From Eq. 4 and Eq.5, the damping coefficients is proportional to the damping cross-section,

$$b = \frac{16N}{3} \sqrt{\frac{\mu k_B T}{2\pi}} \Omega_D$$

where b is the damping coefficients, Ω_D is the damping cross-section of the ions, k_B is Boltzmann constant, μ is the reduced mass of ions and collision gas, T is temperature, and N is the density of ions. Please see the Methods of manuscript on Page 10.

The related equation has been added in the main text on Page 10 Line 202:

“Therefore, the damping coefficient b is proportional to the damping cross-section from Eq. 4 and Eq.5:

$$b = \frac{16N}{3} \sqrt{\frac{\mu k_B T}{2\pi}} \Omega_D \tag{6}”$$

(7) In Figure 1, resolutions on the order of 10,000 are reported, but the sizes (masses and cross-sections) of the three ions are not given.

Re: We have revised the manuscript on Page 4 Line 79 as follows:

“(D) Ion cloud profiling spectra of three biomolecules, lactose (m/z 365, CCS 177.6 Å²), phosphatidylcholine (PC) 18:1/16:0 (m/z 761, CCS 296.2 Å²), and an acetylated peptide (m/z 542, CCS 357.9 Å²), superimposed in one spectrum. The CCS values of lactose and peptide are measured by timsTOF (Bruker Daltonics, Bremen, Germany). The CCS values of phosphatidylcholine is taken from Groessl’s work⁴²”

- (8) It is not clear how the AC excitation is applied and ramped in order to cause selective ejection of the three ions in Figure 1. What is the amplitude and gradient? These are the most important features/parameters and should be included in the main text (and cut the unnecessary common-knowledge information about lipid and glycan isomers). In general, there is key information included in Supporting that is not mentioned in the main text, and much of it is more important than some of the information that is included in the main text.

Re: The related parameters have been added in the main text on Page 3 Line 61:

“For experimental implementations, an auxiliary alternative-current (AC) was employed for resonance excitation of the isomeric ions in the LIT; by scanning the AC voltages in different scan rates depending on the species, the isomeric ions with different DCSs were ejected sequentially according to the ion cloud profile sizes (Fig. 1C and S2). When the AC scanning speed was below 5000 mV/s, a good correlation between AC ejection voltages and DCSs could be established and an ion cloud profiling spectrum with a resolution over 10000 (Fig. 1D) was produced. More specifically, the AC scanning speed was 53 mV/s for the analysis of disaccharides, and 88 mV/s for both phospholipids and peptides (Fig. 1D).”

- (9) Some of the paragraphs are padded with unnecessary facts that are common knowledge, such as “The disaccharides are composed of basic building blocks, the monosaccharides. Each monosaccharide contains multiple hydroxyl groups, which could be connected to form a glycosidic bond with another monosaccharide. Through linking different hydroxyl groups, carbohydrates normally have branched structures with diverse regiochemistry. In addition, each glycosidic bond formation is accompanied by the creation of a stereocenter, because two monosaccharides could be connected via two different configurations.” The same unnecessary padding occurs in the description of phospholipids and peptides.

Re: We have deleted the unnecessary description about disaccharides on Page 5, line 97, and the manuscript has been revised as suggested, please see Page 1 Line 23:

“The significance in identifying isomers is well recognized for almost all disciplines¹⁶. As one example, isomers as cis or trans unsaturated fatty acids in food can have very different effects on human health¹⁷.”

(10) The term “profiling spectra” is too vague. Please consistently use the term “ion cloud profiling spectra”. Is this true or are the ion cloud profiling spectra not displayed in an accurate manner?

Re: We have made revisions as suggested throughout the manuscript and Supplementary Information.

(11) It appears that the ion cloud profiling spectra do not retain quantitative information about the composition of the mixtures. Is this true or are the ion cloud profiling spectra not displayed in an accurate manner?

Re: We have added the quantitative information of different isomers/isoforms of biomolecules in Fig. 2E and Fig. 2F, as shown below:

“Fig. 2: Structural analysis of glycans.

(A) Structures of four isomeric disaccharides, which have isomerization of composition, connectivity, and configuration in pairs. **(B)** Ion cloud profiling spectra of the four disaccharides: trehalose (red), maltose (orange), cellose (green), lactose (blue), and the mixture of these four (black). **(C)** Profiling spectrum of lactose and cellose mixture. **(D)** Tandem MS spectra of lactose and cellose mixture (top), pure lactose (middle), and pure trehalose (bottom). Lactose (blue) and

cellose (green) have identical mass to charge ratio, m/z 365, and fragment, m/z 305, in tandem MS spectra. (E) Calibration curves for pure lactose and cellose. For quantitative analysis of pure sample, concentrations varied from 5 μ M to 50 μ M. (F) Calibration curves for the mixture of lactose and cellose. For quantitative analysis of mixture, concentration of cellose was 5 μ M and concentration ratios of lactose to cellose varied from 0.5 to 10. ”

(12) It is suggested that the ion cloud profiling method is a new separation method. Are the ions truly separated, meaning that they could be captured as separate populations? Or are they simply detected as separated populations?

Re: The separation occurred at the stages of the AC excitation and we used tandem MS spectra to prove ions were truly separated (Figure S4). The disaccharide mixture was trapped in trap 1 and one of the disaccharides was separated to trap 2. Then, the tandem MS spectra were acquired. The characteristic fragments of tandem MS spectra confirmed the separation results.

“Fig. S4. Structural analysis of disaccharide mixture. (A) Ion cloud profiling spectra of trehalose and lactose mixture. (B) Tandem MS spectra of lactose and trehalose mixture (top) and selected lactose (middle) and trehalose ions (bottom) from the mixture using ion cloud profiling. Trehalose (red) and lactose (blue) have characteristic fragments of 203 (red) and 305 (blue), respectively. Trap I was used to select and separate isomers to Trap II and Trap II was used for fragmentation of selected isomers. The excitation energies for ion fragmentation were 180 mV for mixture (top) and 148 mV for both selected lactose (middle) and trehalose ions (bottom), respectively.”

(13) This paper is not cited which previously has shown that CCS information can be determined using an ion trap. (<https://pubs.acs.org/doi/full/10.1007/s13361-017-1720-1>)

Re: We have added a new reference as follows:

31 Dziekonski, E. T., Johnson, J. T., Lee, K. W. & McLuckey, S. A. Determination of collision cross sections using a Fourier transform electrostatic linear ion trap mass spectrometer. *Journal of The American Society for Mass Spectrometry* **29**, 242-250 (2018).

Please see Page 3 Line 41 of the manuscript:

“Typically, IM is performed under a condition $E/N < 30$ Td (Townsend number, $1 \text{ Td} = 10^{21} \text{ V}\cdot\text{m}^2$)²⁸, such as for implementations with drift time²², traveling wave²⁹, trapped³⁰⁻³¹, and differential modes³².”

(14) The impact of the paper would be improved significantly if a correlation can be made between the scanning voltages that are applied and cross-section of the species.

Re: Combining Eq. 2, Eq.6 and Eq. 8 in manuscript in Page 11, we obtain the relationship that the required scanning AC voltage is proportional to the damping cross-section, and it has

$$V_{AC} = \frac{32N}{3} \sqrt{\frac{\mu k_B T}{2\pi}} \alpha^{-1} r_0^2 \omega_0 \Omega_D$$

where V_{AC} is the AC voltage, Ω_D is the damping cross-section of the ions, k_B is Boltzmann constant, μ is the reduced mass of ions and collision gas, T is temperature, and N is the density

of ions, α is calibration coefficient, r_0 is the maximum displacement of ions, ω_0 is the resonance frequency.

The related equation has been added in the main text on Page 11 Line 217:

“Especially, if the displacement of the ion motion is equal to r_0 , we can obtain the correlation between AC ejection voltage and DCS Ω_D , which has

$$V_{AC} = \frac{32N}{3} \sqrt{\frac{\mu k_B T}{2\pi}} \alpha^{-1} r_0^2 \omega_0 \Omega_D \quad (9)$$

The DCS of ions is proportional to the AC ejection voltage. And the larger the DCS is, the greater the AC ejection voltage is required.”

(15) Do the voltages used have any correlation to the actual size of the species? For Figure 3F and 3G: both modifications are on the same amino acid and have a similar sized modification- why is the Vac for the K3 acetylation so low in comparison to the Vac for K9 acetylation and K9/K3 methylation? There doesn't seem to be a distinct correlation of molecular size and Voltage (especially in comparison with Figure 2 displaying Vac for sugars)

Re: The voltages certainly indicate the differences in the size of the species. However, the actual correlations are dependent on the methods used for mobility measurements and may not be linear (especially with high E field, Ref 32) or have the same trends. This could be seen for difference between drift tube and FAIMS. For the specific case here for acetylation, we have verified experimentally the reproducibility of the spectrum. At this moment we cannot yet provide an explanation for the exact difference in comparison with methylation. We plan to continue investigation on this point.

(16) Figure S8 indicates that ion cloud profiling can be used to separate the isomers, select and fragment only one of the isomers, and then detect the unique fragmentation pattern. In the dual ion trap, they state that ion cloud profiling is conducted in the second/back ion trap so it is not clear how the ions could be ejected from this trap to generate the separation but also fragment just one species and then detect it.

Re: The IM separation can be performed using either LIT I or II. For the experiment mentioned here, the mixture of isomers was trapped in Trap I and the AC excitation was applied to separate one of the isomers to Trap II. Trap II was used for the fragmentation of separated isomers. The information has been added in Fig. S4 (Fig. S8 of the original manuscript) for clarification:

“Fig. S4. Structural analysis of disaccharide mixture. (A) Ion cloud profiling spectra of trehalose and lactose mixture. (B) Tandem MS spectra of lactose and trehalose mixture (top) and selected lactose (middle) and trehalose ions (bottom) from the mixture using ion cloud profiling. Trehalose (red) and lactose (blue) have characteristic fragments of 203 (red) and 305, 347 (blue), respectively. Trap I was used to select and separate isomers to Trap II and Trap II was used for fragmentation of selected isomers”

(17) Does the ‘resolution’ of the separation depend on the ion detector used? Would a faster detector offer a higher resolution?

Re: For IM technique, typically the detector response speed is not the limiting factor for the resolution. In this work, the resolution was improved due to the increased tightness of the ion cloud for a same isomer; and the ion cloud density is relatively low to avoid the space charge. The instrument operation status is not approaching the detector response speed limit.

We have revised the manuscript on Page 9 Line 181 as follows:

“For IM technique, typically the detector response speed is not the limiting factor for the resolution and the instrument operation status is not approaching the detector response speed limit.”

(18) All species studied within this work appear to be of single ‘conformation’. It would be worthwhile to show data where a typical mobiligram may show multiple peaks present and how this new ion cloud profiling may distinguish these peaks as opposed to just showing pure isomers and mixtures of them. Many small peptides have various conformations and would be sufficient for this work to prove that distinct features or conformations are observed with this method.

Re: We have analyzed protein ions with different conformations and now data are added for +9 and +13 charge states of myoglobin as well as +12 and +13 charge states of cytochrome C in Fig. S6. In comparison with data obtained previously using other methods, the overlapped peaks now are baseline-resolved, with a good capacity.

“Fig. S6. Ion cloud profiling spectra of proteins with different charge states. Ion cloud profiling spectra of (A) myoglobin +9 charge state, (B) myoglobin +13 charge state, (C) cytochrome c +12 charge state, (D) cytochrome c +13 charge state. Multiple peaks represent different conformations of the protein ions. The results of the conformations of these protein ions are consistent with the previous literatures⁵⁰.”

Data obtained using drift tube IMS (from ref 50)

CCS for (A) myoglobin and (B) cytochrome c. Taken from ref (50).

(19) They mention the potential for use with a TOF or Orbitrap but provide no further information. This seems speculative at best.

Re: This is a speculation, but pointing out an important direction for consideration for the next step in development of IM-MS. Ion trap has been used as an ion processor for high resolution mass analyzer and it should be reasonable to consider of including the IM technology for the instrumentation development. We suggest that this comment be kept in the conclusion.

Reviewer #2 (Remarks to the Author):

This is a very interesting paper that reminds me of a 1999 paper from Graham Cooks that involved seeing “chemical shifts” for isomers analyzed by ion trap MS. The authors have introduced an interesting change of instrumental geometry (compared with Cooks), added off axis fields to turn it into a high resolution technique, and with these changes report a resolving power of 10,000. This is truly spectacular and is enabling for some types of systems – as the authors have demonstrated! The approach that they have taken would allow them to build a library isomers and by scanning in the extra dimension characterize at least some types (and perhaps many types) of isomers. This cannot be done by MS alone. I would not refer to this as a mobility technique; rather I believe they are taking advantage of collisional dampening, which also depends upon the shape. Whatever it is called, this approach will very likely be used broadly for many types of problems. It is important in the field of MS.

I do have some concerns and suggestions for improvement.

- (1)** My take is that the authors have not resolved any pair of ions that cannot be done with existing technology. That is, they resolve species that I believe can be resolved by traditional IM-MS methods and these methods now have quite high resolving powers. See the SLIMS work, circular drift tube and tims work, all commercially available). I have not checked this carefully. But, it would be very straightforward for the authors to choose a system that we know has many structures that cannot be resolved. That is, does the resolving power of the technique allow them to resolve two different chemical species, which differ in structure by 0.01%. Showing that these can be resolved would give the technique a substantial advantage. Some of the reported spectra show a baseline that makes me suspect that these types of ions cannot be separated with this approach. That is, if closely related species were resolvable they should see peaks rather than a continuous baseline.

Re: In the revised manuscript, we have added data for analysis of protein ions with different conformations. Spectra for +9 and +13 charge state of myoglobin, +12 and +13 charge states of cytochrome C are now added as Fig. S6. In comparison with the data obtained using other methods (also attached below, from Ref 50), the overlapped peaks now can be baseline-resolved, and with a good capacity.

“Fig. S6. Ion cloud profiling spectra of proteins with different charge states. Ion cloud profiling spectra of (A) myoglobin +9 charge state, (B) myoglobin +13 charge state, (C) cytochrome c +12 charge state, (D) cytochrome c +13 charge state. Multiple peaks represent different conformations of the protein ions. The results of the conformations of these protein ions are consistent with the previous literatures⁵⁰.”

Data obtained using drift tube IMS (from ref 50)

CCS for (A) myoglobin and (B) cytochrome c. Taken from ref (50).

- (2) Another very exciting aspect of the work is associated with molecular conformation. I would expect the authors' ions to show multiple structures; but, again they are not resolving any differences. This suggests that the dampening essentially anneals the ions to a common structure? Or perhaps the ions remain extremely hot and they have measured an average of all structures for very hot ions? They might add another ionizing couterion – perhaps Ca^{2+} to trap these; or, they could examine a system that is known to display different geometric configurations.

Re: As mentioned above, we have now shown the analysis of protein ions of different conformations in Fig. S6

“Fig. S6. Ion cloud profiling spectra of proteins with different charge states. Ion cloud profiling spectra of (A) myoglobin +9 charge state, (B) myoglobin +13 charge state, (C) cytochrome c +12 charge state, (D) cytochrome c +13 charge state. Multiple peaks represent different conformations of the protein ions. The results of the conformations of these protein ions are consistent with the previous literatures⁵⁰.”

I believe this work will have a substantial impact wherever it is published. However, I'm not sure that it meets the level of uniqueness and includes a new discovery that are typically required to exceed the high threshold associated with Nature.

Re: This work is now being consider by Nature Communications, not Nature anymore.

Reviewer #3 (Remarks to the Author):

Publish after Minor Revisions

The work from Ouyang and co-workers is of high rigor and novelty and will have major impact across analytical and life sciences. Their proposed ion mobility platform, using an easily accessible ion trap at high electric fields, represents a substantial progress in ion mobility instrument development that can be applied to separate isomers of biomolecules.

This said, in order for this work to be appropriate for the broad audience of Nature Communications, the impact needs to be better framed. I recommend publication after the following comments have been addressed:

- (1)** Introduction (lines: 17 – 29): The existence of different isomers and isoforms have been described here and throughout the manuscript, perhaps in a slightly repetitive way. What is missing is a rationale for why the separation of isomers/isoforms is so crucial and what the implications are. The readership of Nature Communications is broad and this is not necessarily obvious for readers from other fields. Here it would be helpful to describe a specific, and broadly accessible example with potential pharmaceutical implications, where commercial state of the art technology reached its limits. It is crucial to describe this impact more clearly in order for the paper to be suitable for this general journal.

Re: We have revised the manuscript on Page 1 Line 23:

“The significance in identifying isomers is well recognized for almost all disciplines. As one example, isomers such as cis or trans unsaturated fatty acids in food can have very different effects on human health.”

- (2)** Line 40: It would be helpful to add that these are normally “low” electric fields.

Re: We have made revisions to the statement from “*condition $E/N < 30 Td$* ” to “*low electric field $E/N < 30 Td$* ” on Page 2 Line 41.

- (3)** IM-MS directly measures the properties of molecular ions and not *just* molecules, please correct that to avoid any misunderstanding for readers not from the field (lines 32, 33, 36).

Re: We have made revisions to the statement from “molecules” to “molecular ions” on lines 32,33,36. Please see Lines 29, 30, and 36 in the manuscript:

“While mass spectrometry (MS) is often used to obtain the compositions of **molecular ions**, ion mobility (IM) has been widely employed for differentiation of the isomers and isoforms of **biomolecular ions** ..., to identify the differences in collision cross-section (CCS) due to the variations in structures of **molecular ions**”

(4) Figure 1: C) Suggest changing the color scheme for readability D) It would be preferred to see a spectrum of the two simulated ions described in B) and C), if possible also simulated. The panel that is now D) could go to E).

Re: Three molecules, lactose, PC 18:1/16:0 and acetylated peptide, in Fig. 1D weren't isomers. So, we added simulation results of four isomeric glycans instead. Please see the Fig. S3 in below:

Fig S3. Experiments and simulation of the four disaccharides isomers. (A) Ion cloud profiling spectra of the four disaccharides: trehalose (red), maltose (orange), cellose (green), lactose (blue). **(B)** Simulated ion trajectories for four isomeric ions characterized by the reduced damping coefficients: $b'_{lactose} = 0.001875$ (blue), $b'_{cellose} = 0.002053$ (green), $b'_{maltose} = 0.002556$ (orange) and $b'_{trehalose} = 0.003004$ (red). Here, $b' = 2b/\Omega m$, where Ω is the angular frequency of the RF field, m is ion mass, b is the damping coefficient of the ions.”

(5) Figure S8: Typo in the caption, it is 347 instead of 307.

Re: We have made a revision to the typo on Page 8 Line 136 in the Supplementary Information. Here shows the main text:

“Trehalose (red) and lactose (blue) have characteristic fragments of 203 (red) and 305, 347 (blue), respectively.”

(6) Please comment on why the $m/z = 347$ peak is not visible in the MS2 spectrum of lactose. As it is presented now, the comparability is limited.

Re: Relative intensities of m/z 305 (through loss of $C_2H_4O_2$) and m/z 347 (through loss of H_2O) are dependent on the CID excitation energy and the number density of the ion cloud, which ultimately also impacts on the excitation during the CID. The intensity of m/z 347 is relatively low at higher excitation energy. A spectrum recorded under this condition is now used instead for Figure S4B upper panel; and the CID excitation conditions are also provided in caption for Figure S4B.

“Fig. S4. Structural analysis of disaccharide mixture. (A) Ion cloud profiling spectra of trehalose and lactose mixture. (B) Tandem MS spectra of lactose and trehalose mixture (top) and selected lactose (middle) and trehalose ions (bottom) from the mixture using ion cloud profiling. Trehalose (red) and lactose (blue) have characteristic fragments of 203 (red) and 305 (blue), respectively. Trap I was used to select and separate isomers to Trap II and Trap II was used for fragmentation of selected isomers. The excitation energies for ion fragmentation were 180 mV for mixture (top) and 148 mV for both selected lactose (middle) and trehalose ions (bottom), respectively.”

(7) Figure S9: I suggest this, or something similar, should appear and be described in more detail in the main paper. The ability to quantify different isomers/isoforms of biomolecules in complex mixtures using IM-MS is of high impact. While the principle is illustrated in Figure S9, it would be good to extend this and e.g. quantify the precision with which the mixture can be analyzed.

Re: The quantitative information has been shown in Fig. 2E and 2F, the R^2 here was determined as 0.9962

“Fig. 2: Structural analysis of glycans.

(A) Structures of four isomeric disaccharides, which have isomerization of composition, connectivity, and configuration in pairs. (B) Ion cloud profiling spectra of the four disaccharides:

trehalose (red), maltose (orange), cellose (green), lactose (blue), and the mixture of these four (black). (C) Profiling spectrum of lactose and cellose mixture. (D) Tandem MS spectra of lactose and cellose mixture (top), pure lactose (middle), and pure trehalose (bottom). Lactose (blue) and cellose (green) have identical mass to charge ratio, m/z 365, and fragment, m/z 305, in tandem MS spectra. (E) Calibration curves for pure lactose and cellose. For quantitative analysis of pure sample, concentrations varied from 5 μM to 50 μM . (F) Calibration curves for the mixture of lactose and cellose. For quantitative analysis of mixture, concentration of cellose was 5 μM and concentration ratios of lactose to cellose varied from 0.5 to 10. ”

- (8) It would be helpful to have an overview table in the main paper, where key properties of your platform (type of IM separation, resolution, cost?, usability, CCS determination possible yes/no etc., similar to Table 1 in 10.1007/s00216-019-01807-0) are compared with both state of the art commercial instruments (please report the highest resolutions known so far for the instruments, e.g. different for Cyclic from the originally reported ones), but also possibly with home-built instruments. Is the reported ion mobility platform the one with the highest resolution so far? This is not clear from the manuscript and needs to be commented on.

Re: Table S1 has been added as a suggestion.

“Table. S1. The key properties of our ion mobility platform and the commercial instruments.

Instruments	Measurable quantities	Calibration required	Resolving powers ($\Omega/\Delta\Omega$)	Acquisition per measurement	Time for a full spectrum
Drift tube	Atmospheric pressure Mobility and CCS (Ω)	No (reference standard)	250 ^[44]	Full spectrum	Milliseconds
IMS	Low pressure Mobility and CCS(Ω); $\alpha(E/N)$		140 ^[45]	Single point	

Cyclotron	Mobility and CCS (Ω)	No	1040 ^[46]	Full spectrum for $n = 1$ Partial spectrum for $n > 1$	Seconds to minutes
Cyclic-/SLIM-TW-IMS	Mobility and CCS (Ω)	Yes	1860 ^[33]	Full spectrum	Milliseconds to seconds
Trapped IMS	Mobility and CCS (Ω)	Yes	400 ^[47]	Single point	Milliseconds to seconds
FAIMS	$\alpha(E/N)$	No ion mobility measurement	460 in $CV/\Delta CV$ ^[48]	Full spectrum	Seconds to minutes
Ion cloud profiling technology	DCS	No typical CCS measurement	10000 in $V_{AC}/\Delta V_{AC}$	Full spectrum	Seconds

Five corresponding references have been added as follows:

- 44 Kirk, A. T., Raddatz, C.-R. & Zimmermann, S. Separation of Isotopologues in Ultra-High-Resolution Ion Mobility Spectrometry. *Anal. Chem.* **89**, 1509–1515 (2017).
- 45 Kirk, A. T., Grube, D., Kobelt, T., Wendt, C. & Zimmermann, S. High-Resolution High Kinetic Energy Ion Mobility Spectrometer Based on a Low-Discrimination Tristate Ion Shutter. *Anal. Chem.* **90**, 5603–5611 (2018).
- 46 Glaskin, R. S., Ewing, M. A. & Clemmer, D. E. Ion Trapping for Ion Mobility Spectrometry Measurements in a Cyclical Drift Tube. *Anal. Chem.* **85**, 7003–7008 (2013).
- 47 Adams, K. J., Montero, D., Aga, D. & Fernandez-Lima, F. Isomer separation of polybrominated diphenyl ether metabolites using nanoESI-TIMS-MS. *Int. J. Ion Mobil. Spectrom.* **19**, 69–76 (2016).
- 48 Shvartsburg, A. A. *et al.* High-Definition Differential Ion Mobility Spectrometry with Resolving Power up to 500. *J. Am. Soc. Mass Spectrom.* **24**, 109–114 (2013).
- (9) Possible Missing Citations in the introduction. (10.1002/mas.21585, 10.1007/s13361-019-02288-2, 10.1007/s00216-019-01807-0)

Re: We have added three new references as follows:

- 25 Dodds, J. N. & Baker, E. S. Ion mobility spectrometry: fundamental concepts, instrumentation, applications, and the road ahead. *Journal of The American Society for Mass Spectrometry* **30**, 2185-2195 (2019).
- 26 Gabelica, V. *et al.* Recommendations for reporting ion mobility Mass Spectrometry measurements. *Mass Spectrometry Reviews* **38**, 291-320 (2019).
- 27 Kirk, A. T. *et al.* Ultra-high-resolution ion mobility spectrometry—current instrumentation, limitations, and future developments. *Analytical and Bioanalytical Chemistry* **411**, 6229-6246 (2019).

Please see Page 2 line 33 of the manuscript:

“Taking the most classic method with drift-tube ion mobility as an example, it uses ion-neutral collisions in an electric field (with a field strength E) to identify the differences in collision cross-section (CCS), which is associated with the variations in structures of molecular ions. Applications of IM have led to the separation of the isomers or isoforms for disaccharides, phospholipids, and peptides with post-translational modifications²⁵⁻²⁷”

(10) The SI Figures are not in the order as they appear in the manuscript.

Re: Checked and corrected.

(11) There are a few typos and phrases where readability could be improved (lines 17, 37, 48, 78, 147). Line 48, Here a new paragraph should start.

Re: Revised as suggestion.

Some for example:

“Biomolecules, such as glycans, lipids, and peptides, play vital roles in biological systems.”
(Page 1 Line 18)

*“Applications of IM have led to the separation of the isomers or **isoforms** for disaccharides, phospholipids, and peptides with post-translational modifications.”* (Page 2 Line 36)

“(new paragraph) Here, we report the development of ion cloud profiling technology to enable high-resolution IM analysis in a radiofrequency (RF) electromagnetic field with $E/N > 1$ MTd.” (Page 3 Line 51)

“Four disaccharides, including trehalose, maltose, cellose, and lactose, were considered.”

(Page 5 Line 89)

REVIEWERS' COMMENTS

Reviewer #1 (Remarks to the Author):

The authors have address all comments (and it should be noted that all three reviewers offered significant feedback, so the level of revisions and new materials are substantial). The revised paper is stronger and will have higher impact. Some of the new additions are impressive. I recommend publication without further review.